# Landslides triggered by the 2015 Mw 6.0 Sabah (Malaysia) earthquake: inventory and ESI-07 intensity assignment

Maria Francesca Ferrario[1]

5 [1] Università dell'Insubria, Dipartimento di Scienza e Alta Tecnologia, Como 22100, Italy

*Correspondence to*: M. F. Ferrario (francesca.ferrario@uninsubria.it)

**Abstract.** On 4 June 2015, a Mw 6.0 earthquake occurred in the Sabah region (Malaysia), triggering widespread landslides along the slopes of Mt. Kinabalu. Despite the moderate magnitude, the Sabah earthquake was very efficient in triggering landslides: here I provide an inventory containing 5198 slope movements, mapped in an 810 km$^2$-wide area. I investigate earthquake intensity using the Environmental Seismic Intensity (ESI-07) scale, which is a macroseismic scale based exclusively on earthquake environmental effects. The epicentral ESI-07 intensity is assessed at IX, considering the dimension of the area affected by secondary effects; such figure agrees well with a dataset of global earthquakes.

I estimate the volume of individual landslides using area-volume scaling laws, then I assigned an ESI-07 intensity to each mapped landslide. I document that the selection of a given area-volume relation has a minor influence on the ESI-07 assignment. Then, I compare ESI-07 values to landslide density and areal percentage on a 1-km$^2$ grid; such parameters are widely adopted in the description of earthquake-triggered landslide inventories. I argue that their integration with the ESI-07 scale may provide an effective way to compare earthquake damage on a variety of spatial and temporal scales. The methodological workflow here illustrated is useful in joining the scientific communities dealing with the realization of earthquake-triggered landslide inventories and with ESI-07 assignment; I believe this effort is beneficial for both the communities.

## 1 Introduction

Moderate to strong earthquakes cause widespread damage due to primary effects (i.e., those related to the seismogenic source, which include surface faulting and permanent ground deformation) or due to ground shaking (i.e., related to the passage of seismic waves). Earthquakes often initiate a cascade of effects, which bring different degrees of hazard and worsen the overall damage (Williams et al., 2018; Fan et al., 2019; Quigley et al., 2020). The frequency and impact of disasters is increasing in the last years and this trend is not expected to change in the future; additionally, modern societies are vulnerable due to the complex interdependencies existing among the territory and infrastructure systems (Harrison & Williams, 2016). Cascading events are function of time and space and follow non-linear paths. When hitting critical nodes, they lead to enhanced direct and indirect losses: thus, assessing systemic interdependencies and including cascading effects into simulation tools is crucial for pursuing a more comprehensive knowledge and supporting preparedness, mitigation and recovery measures (Pescaroli & Alexander, 2016; Zuccaro et al., 2018).

Earthquake damage is usually assessed by means of macroseismic intensity, i.e., a classification of effects on humans, the built and the natural environment (Cecić and Musson, 2004). Among the different intensity scales, the Environmental Seismic Intensity (ESI-07) is the only one based exclusively on environmental effects (Michetti et al., 2004, 2007; Serva et al., 2016; Ferrario et al., 2022). Landslides are one of such environmental effects and may be a significant cause of damage and casualties (Marano et al., 2010; Budimir et al., 2014). Inventories of landslides triggered by earthquakes are crucial for hazard analyses and land planning (Keefer, 1984; Harp et al., 2011; Xu, 2015); currently tens of inventories are available for

a variety of territorial and climatic settings (Schmitt et al., 2017; Tanyas et al., 2017). Landslide inventories were usually derived from manual mapping on aerial or satellite images, but in the last years several efforts have been undertaken to automatically map earthquake-triggered landslides (e.g., Burrows et al., 2020; Handwerger et al., 2020); nevertheless, manually-derived inventories are needed to ascertain the validity and accuracy of (semi)-automatic methods. Landslide number, density and areal percentage vary in the affected area and are often analyzed with respect to topography, seismological or geological conditions (e.g., Chang et al., 2021; Fan et al., 2018; Ghaedi Vanani et al., 2021; Papathanassiou et al., 2021; Wang et al., 2019; Xu et al., 2014).

To date, the scientific communities dealing with the building of landslide inventories and with ESI-07 assignment have proceeded on parallel paths with limited mutual interactions. I believe that an enhanced cooperation may benefit each other: modern landslide inventories have a resolution higher than what is usually achieved by studies focused on the ESI-07 scale; on the other hand, the ESI-07 scale enables the comparison of earthquakes damage through time and space.

Here I analyze the Mw 6.0 Sabah (Malaysia) earthquake, occurred on 4 June 2015. First, I build an inventory comprising 5198 landslides; then, I calculate the landslide number density (LND), landslide area percentage (LAP) and ESI-07 intensity on a 1-km$^2$ grid. ESI-07 assignment requires to convert landslide area to volumes: thus, I explore the epistemic uncertainty associated to different scaling relations. I analyze the interdependency of LND, LAP and ESI-07; since it is expected to have a regional validity, the analysis of additional case histories is needed to assess the reliability of empirical regressions and their stability under different territorial settings. The methodological workflow presented here is aimed at strengthening the exchange of information between different scientific communities; outputs will be useful to inform advancements in ground failure models and for land planning and risk assessment.

## 2 Regional setting and the 2015 Sabah earthquake

### 2.1 Seismotectonic setting

Sabah region lies in a complex seismotectonic setting at the junction of Australian plate, Philippine plate and the Sundaland block. Sabah belongs to Malaysia and it is located in the northern part of Borneo Island. Seismicity is diffuse along the plate boundaries (Fig. 1a), where the subduction interface is located. Less frequent earthquakes have been recorded in the Ranau region, including a Mw 5.3 in 1966 and a Mw 5.2 in 1991. Offshore Sabah, the NW Borneo trench is a deep-water fold-and-thrust belt; its structural setting is debated and it has been related either to gravity sliding or to tectonic shortening (Hall, 2013; Sapin et al., 2013). GPS measurements show that, despite the absence of seismicity, the NW Borneo trench may accommodate up to 5 mm/yr (Simons et al., 2007). GPS data also assess that Sabah is actively deforming, albeit at a slower rate than the surroundings (Simons et al., 2007; Mustafar et al., 2017); this contradicts the earlier view of a rigid Sundaland block.

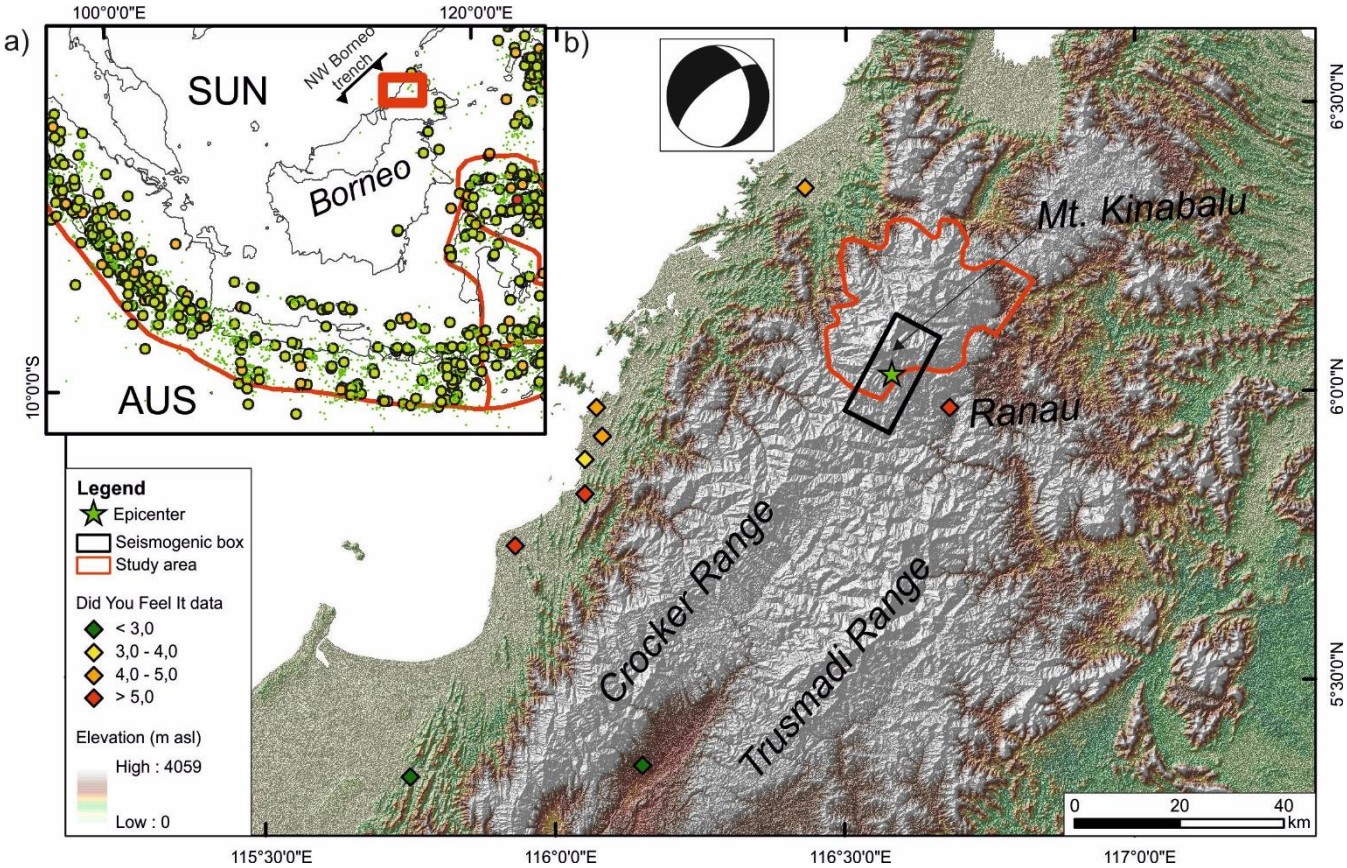

**Figure 1: a) Regional seismotectonic setting of SE Asia showing main plate boundaries and M > 5 earthquakes (USGS/NEIC catalogue); AUS: Australian Plate, SUN: Sundaland block; the red rectangle marks the area enlarged in b. b) Digital elevation model derived from 30-m resolution ALOS DEM; focal mechanism is from USGS, epicenter and seismogenic box after Wang et al. (2017); intensity data from DYFI program are shown as well.**

## 2.2 The study area and the 2015 Sabah earthquake

Sabah is characterized by rugged topography, dominated by the Crocker and Trusmadi Ranges; the highest peak is Mount Kinabalu, reaching 4100 m asl and representing the first World Heritage Site in Malaysia. It is a granitic pluton exposed over a ca. 120 km$^2$-wide area and it was exhumed about 7 My ago (Cottam et al., 2010). Beside the granitic pluton, Oligocene-Lower Miocene sandy turbidites constitutes the Crocker Formation, while the Trusmadi Formation comprises argillite, slate, siltstone and sandstone with volcanics (Hutchinson et al., 2000). Sabah is covered by thick tropical forests and the climate is characterized by monsoonal seasons (November to March and May to September); rainfall is high (> 3000 mm/yr) but highly variable due to local topography (Menier et al., 2017).

Several faults have been mapped in the region, mainly based on tectonic geomorphology and watershed analyses (e.g., Mathew et al., 2016; Menier et al., 2017; Shah et al., 2018). Sedimentary basins bounded by normal faults are indeed aligned

along the Crocker and Trusmadi Ranges; geomorphological features pointing to a recent tectonic activity include triangular facets, scarps and river anomalies (Tija, 2007; Tongkul, 2017; Wang et al., 2017). Laterally offset features (terraces, river courses) allow to identify strike-slip structures that crosses Borneo (Shah et al., 2018).

The Mw 6.0 Sabah earthquake occurred on 4 June 2015 at 23.15 UTC at 10 km depth (USGS, 2018); it is the largest instrumental event in the province. The event had a normal focal mechanism, with a NE-SW oriented main focal plane. The seismogenic box and relocated epicenter after Wang et al. (2017) are shown in Fig. 1b. The seismogenic source of the 2015 Sabah earthquake belongs to a system of normal faults of about 200 km length that lies at the foothills of the Crocker Range (Tjia, 2007; Tongkul, 2016, 2017; Wang et al., 2017).

"Did You Feel It?" data acquired by the USGS include sparse intensity estimations, with maximum values of 6.6 on the CDI (Community Decimal Intensity) scale at Ranau. The earthquake did not generate primary surface faulting and a directivity toward Mt. Kinabalu has been inferred based on teleseismic waveform inversion and space-based geodesy (Wang et al., 2017). The event generated thousands of landslides and rockfalls (Tongkul, 2017; Wang et al., 2017) which caused the death of 18 people along hiking routes on Mt. Kinabalu. Additionally, water infrastructures were damaged and local businesses badly affected (Lehan et al., 2020).

The landslide deposits provided abundant sediments for subsequent remobilization as debris flows following heavy rainfall (highest rainfall intensity of 14.2 mm/h on 15 June 2015; Rosli et al., 2021a). Some detailed studies of debris flows were performed on limited areas through Lidar techniques (Yusoff et al., 2016; Rosli et al., 2021a, b), but a comprehensive inventory of all the triggered landslides is still lacking and is the focus of this paper.

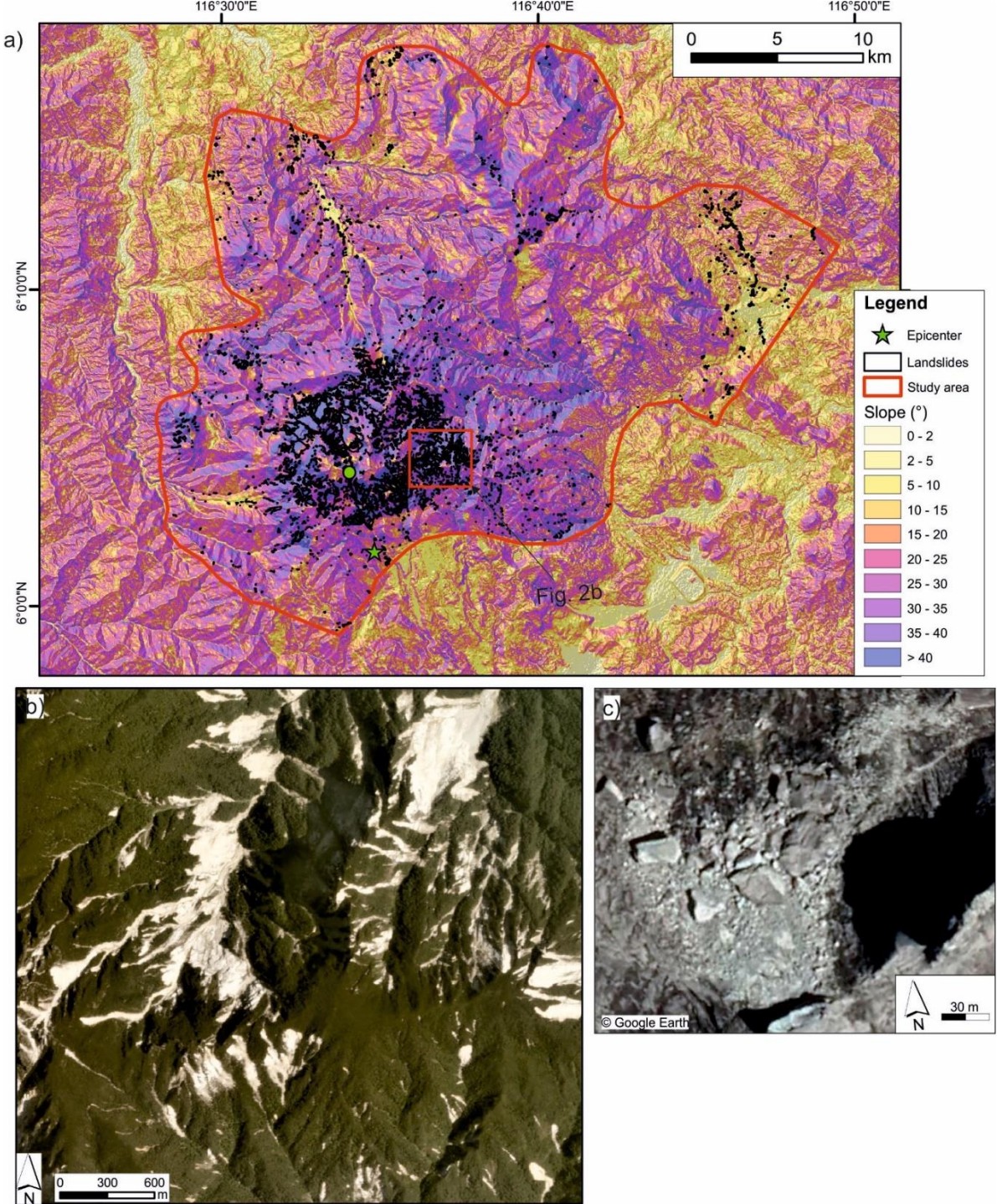

**Figure 2: a) Slope map and hillshade derived from 30-m resolution ALOS DEM; the red polygon is the study area, while landslides are shown in black; b) Planet Labs image (3-m resolution) taken on 18 March 2016 showing widespread landslides; c) © Google Earth image of a rockfall on top of Mt. Kinabalu (location is the green dot in Fig 2a).**

## 3 Materials and methods

### 3.1 Landslide inventory

I realized a landslide inventory in an 810 km$^2$-wide area (red polygon in Fig. 2a) using QGIS software; the inventory is based on visual interpretation of 3-m resolution PlanetScope satellite images. I used high resolution Google Earth images to gain a regional overview, while individual landslides are mapped on ortho-rectified 4-band multispectral tiles. The landslide inventory is realized on images taken on 23 February, 18 and 21 March 2016; such images postdate the earthquake by about 8 months and thus the inventory has to be intended as the cumulative damage due to the mainshock, aftershocks and additional remobilization (e.g., debris flows, Rosli et al., 2021a, b). This is a limitation that should be considered when comparing the obtained database with other case histories; it is due to persistent cloud cover that prevented the acquisition of clear images over the entire area closer in time to the mainshock. Cloud-free PlanetScope images are not available for the time preceding the earthquake; preexisting landslides were thus identified from multi-temporal Google Earth historical images, acquired between May 2008 and April 2015.

Landslides were mapped as polygons encompassing the source and deposit areas. Shallow landslides were easily recognizable in forested regions, since they stripped off the vegetation (Fig. 2b). Mapping was more difficult in the higher part of Mt. Kinabalu pluton, since bare rock was already outcropping before the earthquake; in this sector, brighter colors on post-event images was used as an indication of the occurrence of slope movements. Landslide mapping may suffer from problems related to amalgamation of coalescing polygons, i.e., the mapping of several adjacent landslides as a single polygon (Marc and Hovius, 2015). This problem is especially severe for inventories realized through automatic mapping and may introduce a bias in the computation of landslide number and other statistics (e.g., ESI-07 assessment). When multiple sources areas coalesce in a single toe sector, it is difficult to identify individual landslides. In such cases, I first mapped the entire polygon, then I used the "split" tool to delineate the different source areas. This GIS tool allows to draw contiguous polygons, avoiding the overlap of different polygons, or unmapped areas in between.

### 3.2 Landslide number density (LND), landslide areal percentage (LAP) and ESI-07 intensity assignment

The study area (Fig. 2a) was divided in a grid of 1 km x 1 km cells and the centroids of each landslide polygon were extracted. The LND is calculated as the sum of the centroids fitting in each 1 km x 1 km grid cell. LAP represents the percentage of the area covered by the mapped polygons within each cell. Additionally, I define "landslide area" as the sum of areas of individual landslides, while I use "affected area" to indicate the region encompassing all the mapped slope movements.

ESI-07 intensity assignment requires to estimate volumes of each landslide. This can be achieved via field surveys, which however are not feasible for all the landslide population, or by differencing of high resolution pre- and post-landslide elevation models (e.g., Massey et al., 2020). When such data are not available, area-volume empirical relations are

commonly used (Guzzetti et al., 2009; Fan et al., 2019); to assess the epistemic uncertainty related to the area-volume conversion, I tested multiple equations (Table 1), which have the general form:

$$V = \alpha \times A_i^{\gamma} \quad (1)$$

145 Where V is volume in $m^3$, $A_i$ is the area of individual landslides in $m^2$, and $\alpha$ and $\gamma$ are fitting coefficients.

**Table 1: Area-volume scaling relations considered in this study.**

| Nr. | Equation | α | γ | Notes |
|---|---|---|---|---|
| 1 | Guzzetti et al., 2009 | 0.074 | 1.450 | Global, slide type, several triggering processes |
| 2 | Larsen et al., 2010 (all) | 0.146 | 1.332 | Global, all types |
| 3 | Larsen et al., 2010 (bedrock) | 0.186 | 1.350 | Global, bedrock |
| 4 | Larsen et al., 2010 (soil) | 0.257 | 1.145 | Global, soil |
| 5 | Xu et al., 2016 | 1.315 | 1.208 | Subset of landslides triggered by 2008 Wenchuan earthquake |
| 6 | Benjamin et al., 2018 | 0.588 | 1.202 | Rockfalls on coastal cliffs at Staithes (UK); 2D change detection from Terrestrial Laser Scanner point clouds |
| 7 | Caputo et al., 2018 | 0.729 | 1.125 | Rockfalls on coastal cliffs at Coroglio (Italy); volume estimated from Terrestrial Laser Scanner data |

The ESI-07 guidelines (Michetti et al., 2004; 2007) include typical values of landslide volume for each intensity degree;
150 thus, I used the volume derived with Equation (1) to assign an ESI-07 intensity to each landslide polygon, following the thresholds presented in Table 2. It must be noted that landslide dimension saturates at ESI-07 X (i.e., it is not possible to define degrees higher than X based on individual landslides). To compare ESI-07 to LND and LAP values, the highest ESI-07 value is retained for each grid cell, adopting an approach similar to Ota et al. (2009) and Silva et al. (2013).

LND, LAP and ESI-07 focus on different aspects, as illustrated in Fig. 3. The three panels have the same LAP (36% of the
155 "study region", i.e., the black square); the area of the biggest landslide is used to compute the ESI-07 value, as done for the real case study. In this example, the Guzzetti et al. (2009) equation is used to illustrate the results. The first scenario (Fig. 3a) shows one wide landslide, resulting in an ESI-07 value ≥X. The second case shows 36 small landslides and is equivalent to ESI-07 VIII. The third case shows the presence of one medium-sized and 30 small landslides, resulting in ESI-07 value of IX. Fig. 3 highlights that the concurrent evaluation of LND, LAP and ESI-07 provide an added value in the understanding of
160 the distribution and characteristics of the landslide inventory, due to the role played by the number and dimension of individual landslides in the calculation of the different metrics.

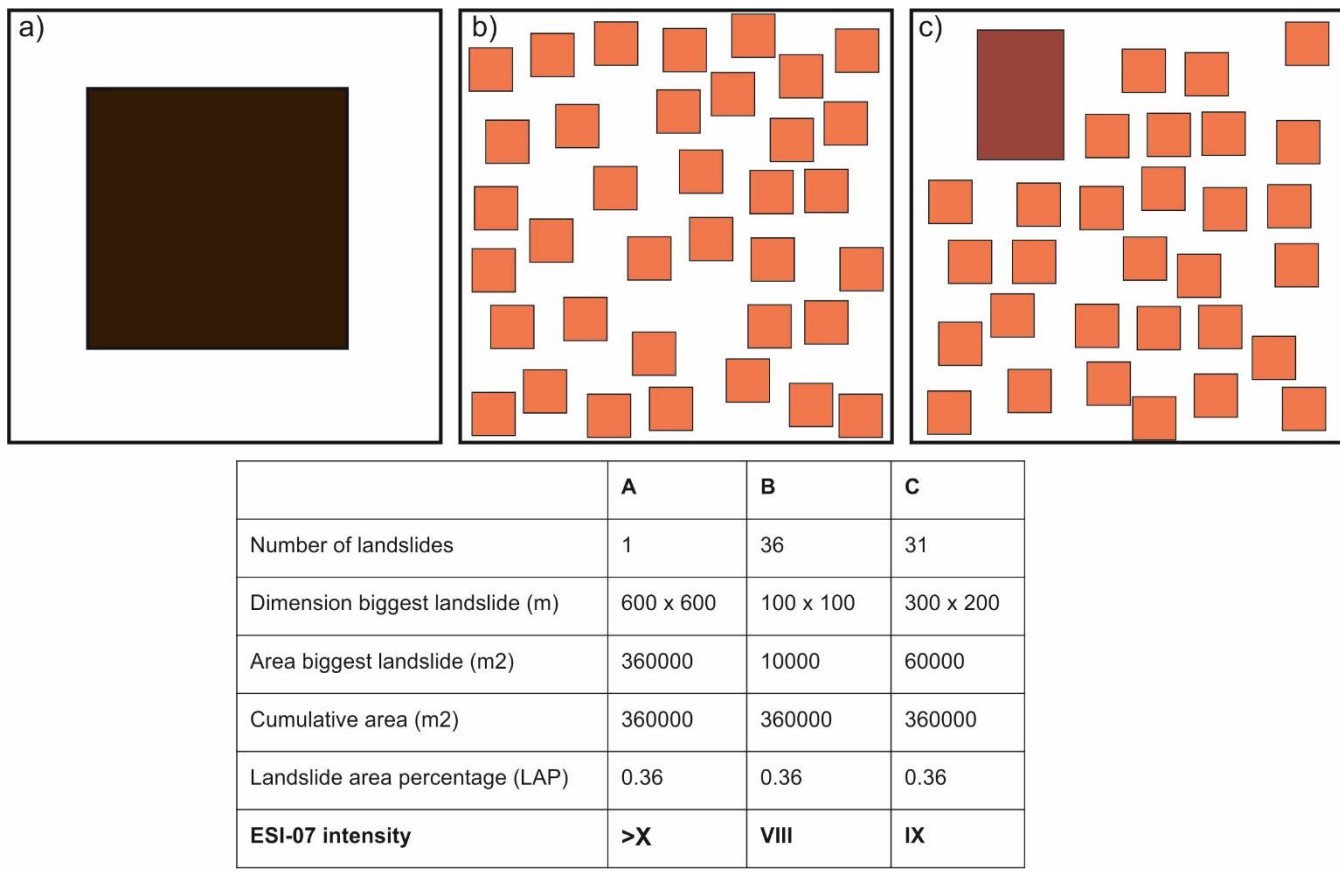

| | A | B | C |
|---|---|---|---|
| Number of landslides | 1 | 36 | 31 |
| Dimension biggest landslide (m) | 600 x 600 | 100 x 100 | 300 x 200 |
| Area biggest landslide (m2) | 360000 | 10000 | 60000 |
| Cumulative area (m2) | 360000 | 360000 | 360000 |
| Landslide area percentage (LAP) | 0.36 | 0.36 | 0.36 |
| **ESI-07 intensity** | **>X** | **VIII** | **IX** |

**Figure 3: Sketch illustrating the role of number and dimension of landslides (coloured squares) on the computation of LND, LAP and ESI-07. The upper panels represent simplified scenarios: a) one wide landslide; b) many landslides, all of small dimension; c) many landslides with variable dimension.**

Finally, the Sabah case study is compared to other landslide inventories on a global scale; in particular, I used the scaling relations of Malamud et al. (2004a, b), which relate the number of triggered landslides, earthquake magnitude and total landslide area:

$$logN = 1.27 \times M - 5.45\ (\pm0.46)\quad(2)$$
$$logA_{LT} = 1.27 \times M - 7.96\ (\pm0.46)\quad(3)$$
$$A_{LT} = 3.07 \times 10^{-3} \times N\quad(4)$$

Where N is the number of landslides, M is moment magnitude and $A_{LT}$ is the total landslide area in km$^2$. Equations (2) and (3) are from Malamud et al. (2004b), while Equation (4) is from Malamud et al. (2004a).

**Table 2: Landslide volumes used in this study to assign ESI-07 local intensities.**

| ESI-07 degree | VI | VII | VIII | IX | X - XII |
|---|---|---|---|---|---|
| Landslide volume (m$^3$) | $<10^3$ | $10^3 - 10^4$ | $10^4 - 10^5$ | $10^5 - 10^6$ | $>10^6$ |

## 4 Results

### 4.1 Spatial distribution of landslides

The inventory for the 2015 Sabah earthquake comprises 5198 landslides mapped in an 810-km$^2$ wide area, thus resulting in an average of 6.4 landslides/km$^2$. Landslides have an average area of 3625 m$^2$. The slope movements are not equally distributed in space, but instead concentrates in a steep zone along the slopes of Mt. Kinabalu (Fig. 2a). Outside the Mt. Kinabalu pluton, landslides clusters in small patches, while the surrounding territory is unaffected. Summing the area of single landslides, a total of 18.84 km$^2$ is obtained, which represents the 2.33% of the investigated area. Landslides are located north of the epicenter (Fig. 2a), possibly reflecting the rupture directivity which enhanced ground shaking in this direction (Wang et al., 2017).

Fig. 4 presents the grid maps of landslide density number (LND) and of landslide area percentage (LAP). Maximum values reach LND = 99 landslides/km$^2$ and LAP = 68%; the mapped area includes 895 cells, but landslides were mapped only in about 67% of the cells (see the distribution of landslides in Fig. 2a). Overall, there is a good agreement between LND and LAP and the spatial distribution of the two descriptors is fairy similar (Fig. 4). The distribution of coseismic landslides can be compared to expected ground failures: the USGS routinely provides such information in the aftermath of strong events, using models based on seismological, topographic and geological variables (Nowicki Jessee et al., 2018). For the Sabah earthquake the model correctly recognizes the slopes of Mt. Kinabalu as the focus of the highest damage and matches fairly well with actual slope movements (Supplementary Fig. S1).

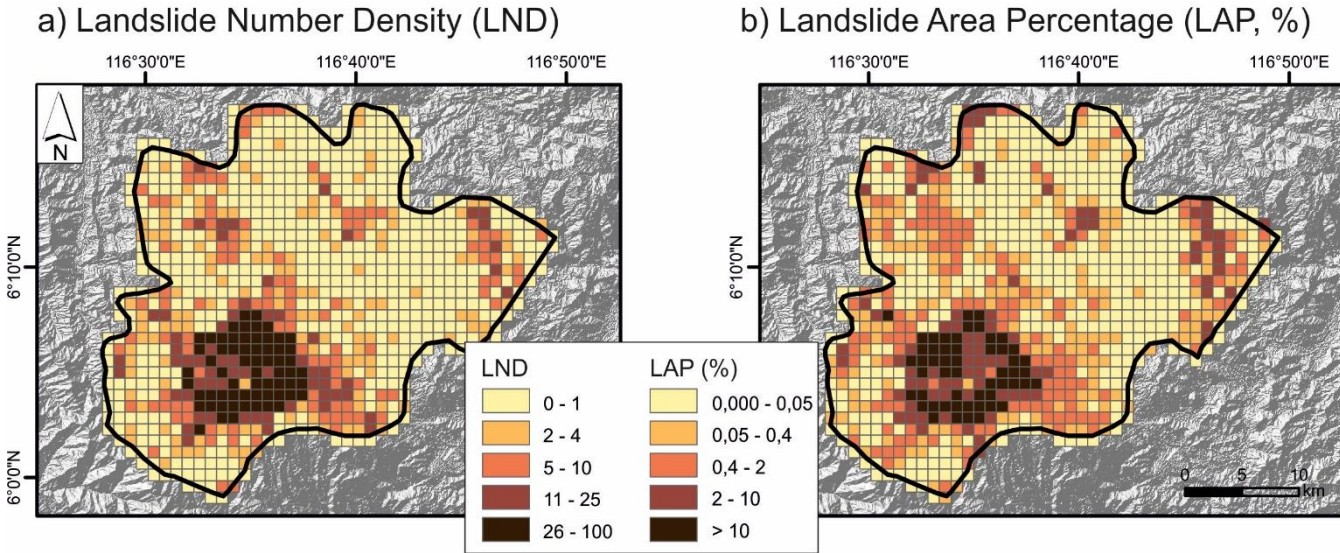

**Figure 4: Grid maps of Landslide Number Density (LND, a) and Landslide Area Percentage (LAP, b). Colormap follows Crameri et al. (2020).**


### 4.2 ESI-07 macroseismic field

I compute the landslide volume using Equation (1); to assess the influence of a given scaling relation, I test seven different models (see Table 1). They encompass different climatic and regional settings and have been derived either from global (Guzzetti et al., 2009; Larsen et al., 2010) or regional (Xu et al., 2016) datasets. One equation (Xu et al., 2016) derives from

earthquake-induced landslides, while the others refer to landslides triggered by multiple processes (i.e., earthquakes, rainfall, snowmelt). Two of the equations (Benjamin et al., 2018; Caputo et al., 2018) specifically deal with rockfalls.

Fig. 5 presents the grid maps for the seven scaling relations: when multiple landslides lie in a single cell (i.e., LND > 1), I retain the highest ESI-07 value. Notwithstanding the selected scaling relation, the spatial distribution of ESI-07 values shows a common pattern; this is further summarized in Fig. 5h, where the number of cells belonging to each ESI-07 intensity class

is shown for the seven relations. The Larsen et al. (2010) "soil" regression results in lower intensities than the other equations, while the Xu et al. (2016) regression provides the highest number of cells with intensity ≥ VIII. The number of cells having an ESI-07 intensity ≥ X ranges from 2 (Larsen et al., 2010 "soil") to 28 (Xu et al., 2016), which represent the 0.2 – 3.1% of the total cells. One limitation of the equations specific for rockfalls (Benjamin et al., 2018 and Caputo et al., 2018) is the dimension of the individual rockfalls, which in both cases do not exceed 30 m$^2$. The extrapolation of the A-V

relations to much bigger landslides should be carefully considered; nevertheless, the 7 relations considered in this study clearly show a similar picture in terms of ESI-07 distribution, testifying that input data (i.e., landslide inventory) are far more important than the choice of the area-volume relation. This is because ESI-07 degrees are based on broad categories in terms

of volume (each category span at least one order of magnitude, Table 2); observed differences between the ESI-07 maps are related to landslides whose volumes are close to the boundaries defined in the ESI-07 scale.


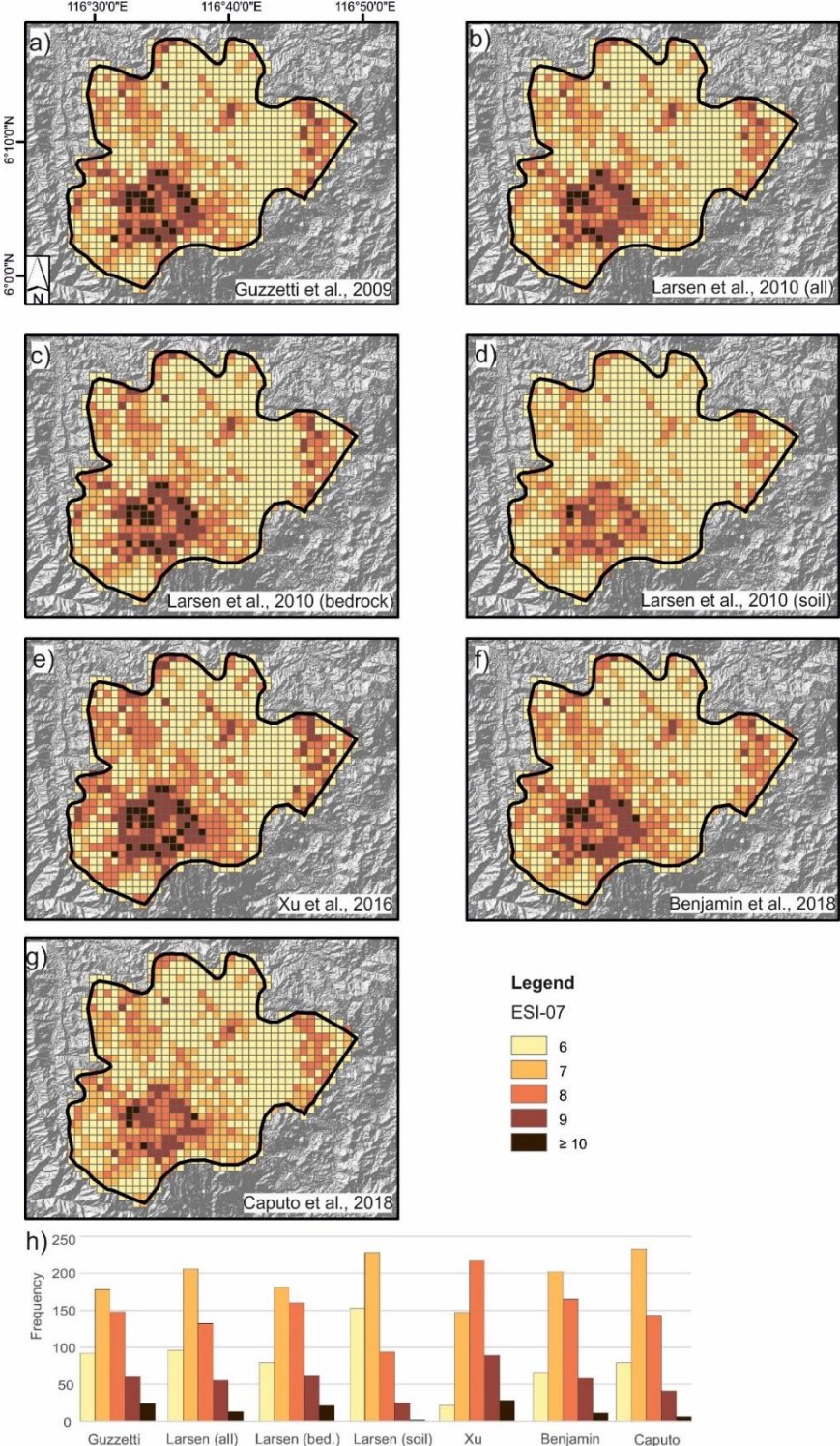

**Figure 5: Grid maps of ESI-07 local intensity obtained by adopting different area-volume scaling relations (a: Guzzetti et al., 2009; b: Larsen et al., 2010 – all types; c: Larsen et al., 2010 – bedrock; d: Larsen et al., 2010 – soil; e: Xu et al., 2016; f: Benjamin et al., 2018; g: Caputo et al., 2018); h: frequency of cells belonging to the different ESI-07 classes for each scaling law.**


## 5 Discussion

### 5.1 Challenges and data limitations

The Sabah earthquake and the methodological approach presented in this study highlight some of the challenges commonly encountered when analyzing earthquake-induced landslide inventories. Indeed, the realization of a reliable landslide inventory requires the fulfilment of several criteria, either in terms of available images or mapping methodology (Harp et al., 2011). In this section, I discuss the role of pre-existing landslides, i.e., slope movements already present before the Sabah earthquake (Section 5.1.1) and other sources of epistemic uncertainty (Section 5.1.2).

### 5.1.1 The role of pre-existing landslides

A co-seismic landslide inventory should include only those slope movements triggered, or reactivated, by the seismic shaking. The inventory presented in this paper is realized on a homogeneous dataset of satellite images provided by PlanetScope; similar images are not available for the timeframe antecedent the earthquake, introducing a difficulty in the evaluation of whether a landslide was already present before the earthquake. Thus, I realized a dataset of pre-existing landslides by inspecting Google Earth historical images, ranging from 19 May 2008 to 2 June 2015. Cloud-free images are not available for a 186 km$^2$ wide region, corresponding to 23% of the total area (blue region in Fig. 6). I mapped a total of 225 pre-existing landslides and compute LND and LAP with the same procedure adopted for the co-seismic inventory. It is evident that pre-existing landslides exert a very limited role, either in terms of total number (225 pre-existing vs 5198 co-seismic) and area (0.55 vs 18.84 km$^2$). In Fig. 6, I adopted the same color scheme as for the co-seismic inventory (Fig. 4), to highlight that more than 95% of the grid cells belong to the lowest LND class (less than 2 landslides per cell), while more than 99% of the cells belong to the lowest LAP class (max LAP value is 0.09%).


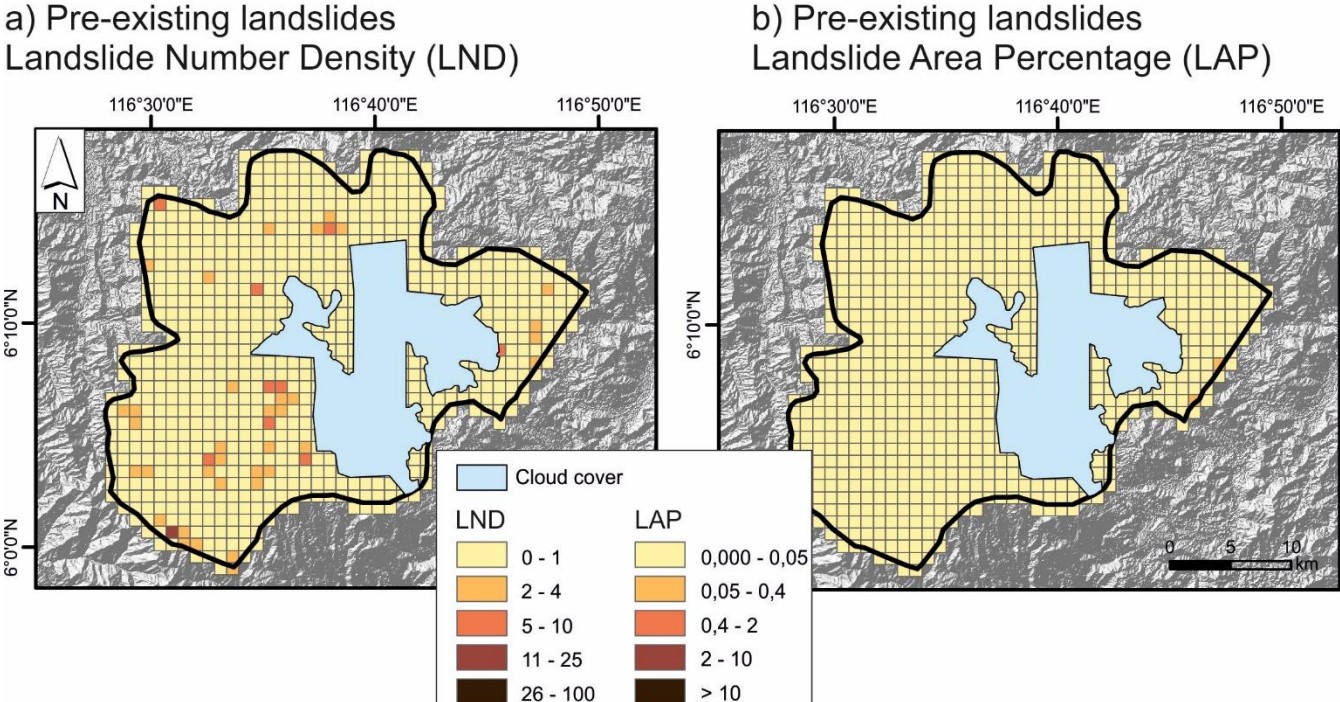

**Figure 6: Grid maps of Landslide Number Density (LND, a) and Landslide Area Percentage (LAP, b) for pre-existing landslides; the blue area represents a region where cloud-free pre-earthquake imagery is lacking. Colormap follows Crameri et al. (2020).**

### 5.1.2 Chain of hazards and other sources of epistemic uncertainty

Several processes can trigger landslides, such as seismic shaking, heavy rainfall, anthropic disturbances. These processes may act concurrently or have complex interdependencies among each other, resulting in a so-called chain of hazards (e.g., Fan et al., 2019). The identification of the precise event that triggered the landslides can be challenging, and subsequent remobilizations may occur as well. In the Sabah case, slope movements were firstly triggered by the seismic shaking; later on, prolonged rainfall reactivated the landslide deposits as debris flows (Rosli et al., 2021a). In this work, landslides were mapped on optical satellite images, whose availability depends on satellite revisit time and local weather conditions. The co-seismic landslide inventory is realized on images acquired about 8 months after the Sabah earthquake, since persistent cloud cover hampered the analysis of a shorter time interval. This point is a significant source of epistemic uncertainty, which is difficult to reduce, unless other data sources are present (e.g., field surveys, helicopter/drone flights).

Chain of hazards affect the territory for prolonged times: the remobilization of deposits results in enhanced rates of slope movements; these processes may take 5-10 years (Avsar et al., 2016) and generate bank erosion or floodplain accretion downstream, thus affecting flood frequency (Fan et al., 2019). Stochastic natural processes (e.g., earthquakes) and seasonal hazards (e.g., rainfall, flood) imply different modeling tools and calls for complex risk reduction strategies (Quigley et al.,

2020); understanding cause-effect relationships and latent vulnerabilities helps in informing such efforts (Pescaroli &
Alexander, 2016). Additionally, landslide phenomena triggered by human activities are increasing and have an influence
comparable, if not higher, than natural processes such as rainfall or earthquakes (Froude & Petley, 2018; Tanyas et al.,
2022). The identification of critical nodes and interdependencies among cascading hazards can be beneficial for the
development of targeted mitigation strategies.

Another source of epistemic uncertainty is related to the area-volume scaling relations adopted to compute ESI-07 values.
Many equations have been proposed in the literature, referring to different triggering processes (e.g., seismic shaking vs
rainfall), climate conditions (specific regions vs global validity), landslide type (slides vs rockfall), mapping procedures
(e.g., landslides delineated as single polygons vs separation of source and deposit area) and methods for data acquisition
(e.g., manual vs automatic mapping; satellite vs drone images vs laser scanner techniques). Thus, the selection of the most
suitable equation may not be straightforward. In Section 4.2 I used seven different equations to derive the ESI-07
macroseismic field; results demonstrate that the epistemic uncertainty related to the choice of the area-volume relation is
much lower than other sources of uncertainty.

## 5.2 Comparing the 2015 Sabah case study with worldwide data

Here I compare the Sabah case study to other landslide inventories on a global scale, to evaluate its characteristics in a
broader context; eventual peculiar characteristics are then discussed. Fig. 7 summarizes the characteristics of a number of
earthquake-triggered landslide inventories, represented as a function of earthquake moment magnitude. Open symbols
represent data collected from published literature; the dataset is available on the Zenodo repository (see Data availability
section). Fig. 7a shows the number of triggered landslides with respect to Mw; Equation (2) and its confidence bounds are
shown as well. The Sabah case history lies well above the expected value, probably because it includes both strictly
earthquake-triggered landslides and material remobilized by subsequent debris flows (Rosli et al., 2021a, b). Fig. 7b shows
the dimension of the area affected by landslides; in this case, the Sabah inventory is in good agreement with global studies
and lies just below the upper bound proposed by Keefer (1984; solid line). Fig. 7c presents the total landslide area (sum of
areas of individual landslides), together with Equation (3) and relative confidence bounds; the Sabah earthquake seems an
outlier in the data population, although the debris flow remobilization may make the landslide area estimate not fully reliable
for the Sabah earthquake. On the contrary, by adopting the relation based on number of landslides (i.e., Equation 4), the
expected landslide area of 15.96 km$^2$ is in fair agreement with the observed value of 18.84 km$^2$. It must be noted that the
works by Keefer (1984) and Malamud et al. (2004b) were based on a subset of the datapoints in Fig. 7; many inventories
were realized in the last few years, possibly arguing for the need of updating the scaling relations. Nevertheless, for a given
Mw the plots show a high variability, spanning about 3 orders of magnitude in terms of number of landslides, affected area
and landslide area. Such behavior is related to inherent variability in landslide occurrence across varying geological settings:
the local conditions play a prominent role in driving secondary earthquake environmental effects (Keefer, 2002; Michetti et

al., 2007; Fan et al., 2019). Finally, Fig. 7d shows the distribution of ESI-07 epicentral intensity as a function of Mw (Ferrario et al., 2022). The ESI-07 epicentral intensity is assigned based either on the dimension of the affected area, or on the dimension of the biggest effects. I assign an ESI-07 epicentral intensity of IX to the Sabah case history: the area encompassing all the mapped landslides is 810-km$^2$ wide, which fits the description in the ESI-07 guidelines (*"the affected area is usually less than 1000 km$^2$"*; Michetti et al., 2004). The Sabah case study is widely in agreement with the dataset.

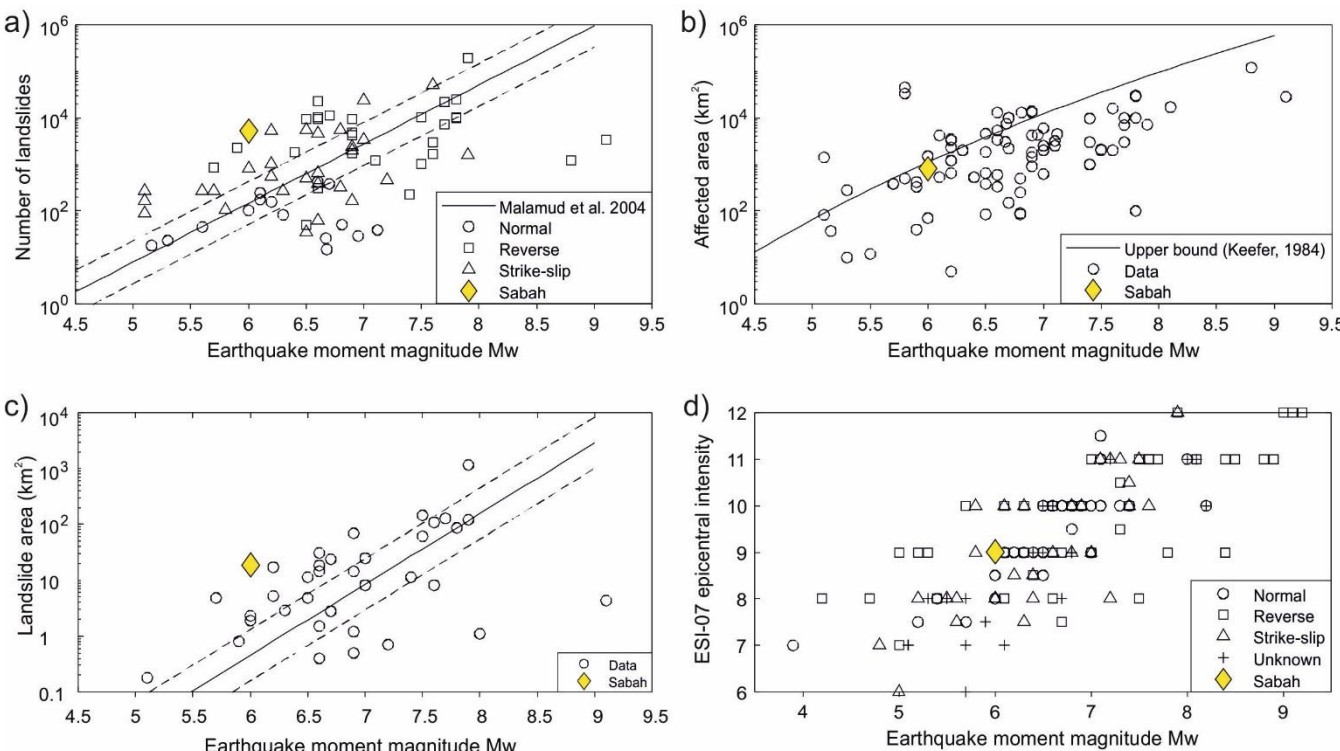

**Figure 7: Comparison of the Sabah earthquake with global studies: a) number of landslides vs Mw, regression is Equation (2); b) affected area vs Mw, upper bound after Keefer, 1984; c) landslide area vs Mw, regression is Equation (3); d) ESI-07 epicentral intensity.**

The Sabah earthquake produced a higher number of slope movements and a higher landslide area (sum of areas of individual landslides) than events of similar magnitude. This fact can be related to two alternative explanations:

- The 2015 earthquake is the strongest event in Sabah in the instrumental era: infrequent strong events may be highly efficient in triggering a large number of landslides.
- I realized the inventory on satellite images acquired 8 months after the earthquake, thus slope movements triggered by processes other than the mainshock may be included, such as debris flow remobilization. This implies that comparison with other earthquakes should be gingerly considered.

### 5.3 Scaling relations among LND, LAP and ESI-07

In Fig. 8 I show the distribution of ESI-07 intensity with respect to LND and LAP values of each grid cell. The graphs refer to the results obtained with the Guzzetti et al. (2009) equation, but a similar picture is obtained when applying the other equations of Table 1. The median LND and LAP values for each ESI-07 intensity class are presented in Table 3 and Table 4: it can be noticed that in some instances (Larsen et al., 2010 "bedrock" and "soil"; Caputo et al., 2018) LND values for the ESI-07 class IX are higher than X, but this inversion is possibly driven by the limited number of cells in the ESI-07 X class. Median LAP values instead do not show such inversions, eventually suggesting that LAP is a better descriptor than LND for assessing the damage. This fact is not surprising, since LND has a "point" validity, while LAP is related to an area assessment, which should generally be more consistent with volume (on which ESI-07 values are based). Additionally, LAP may be more stable than LND with respect to epistemic uncertainty, because the number of mapped landslides (and thus LND) is strongly dependent on the resolution of images used for building the inventory and may be affected by amalgamation issues (Marc & Hovius, 2015).

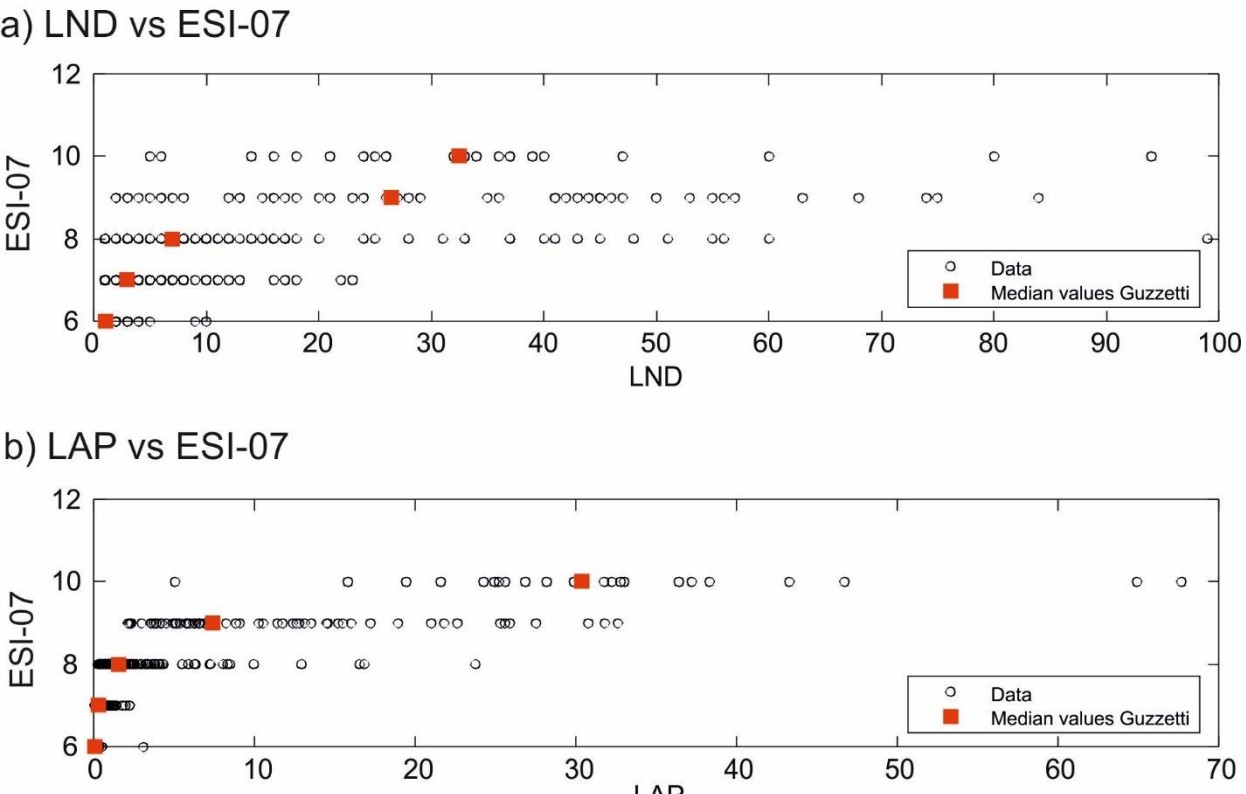

**Figure 8: Plots of LND (a) and LAP (b) vs local ESI-07; intensity computed using the Guzzetti et al. (2009) scaling relation is shown as an example.**

**Table 3: Median values of LND for each ESI-07 intensity degree, obtained using the area-volume scaling relations of Table 1.**

| ESI-07 | Guzzetti | Larsen (all) | Larsen (bedrock) | Larsen (soil) | Xu | Benjamin | Caputo |
|--------|----------|--------------|------------------|---------------|-----|----------|--------|
| VI | 1 | 1 | 1 | 2 | 1 | 1 | 1 |
| VII | 3 | 3 | 3 | 5 | 2 | 2 | 3 |
| VIII | 7 | 8 | 7 | 17 | 5 | 8 | 9 |
| IX | 27 | 29 | 27 | 33 | 17 | 29 | 35 |
| ≥ X | 33 | 26 | 32 | 23 | 33 | 32 | 22 |

**Table 4: Median values of LAP (%) for each ESI-07 intensity degree, obtained using the area-volume scaling relations of Table 1.**

| ESI-07 | Guzzetti | Larsen (all) | Larsen (bedrock) | Larsen (soil) | Xu | Benjamin | Caputo |
|--------|----------|--------------|------------------|---------------|-----|----------|--------|
| VI | 0.05 | 0.05 | 0.04 | 0.09 | 0.023 | 0.04 | 0.04 |
| VII | 0.28 | 0.34 | 0.24 | 0.70 | 0.10 | 0.24 | 0.33 |
| VIII | 1.55 | 1.91 | 1.53 | 5.76 | 0.74 | 1.70 | 2.35 |
| IX | 7.39 | 13.55 | 8.23 | 29.85 | 5.87 | 14.02 | 22.62 |
| ≥ X | 30.35 | 33.02 | 30.35 | 55.48 | 30.10 | 33.02 | 44.99 |

### 5.4 Prospect for future work

LND and LAP have been frequently explored in the realm of earthquake-triggered landslide inventories (e.g., Fan et al., 2018; Ferrario, 2019; Ghaedi Vanani et al., 2021; Xu et al., 2014), while a grid approach has been seldom applied in the assessment of ESI-07 intensity, with the exceptions of Ota et al. (2009) and Silva et al. (2013). The current work is the first attempt toward a quantitative relation among LND/LAP and ESI-07. In Section 5.3, I describe the relations obtained for the Sabah earthquake; nevertheless, reliable empirical relations should be based on a wider dataset and not on a single case history. Scaling relations are indeed expected to have a regional validity and thus it is necessary to investigate the inter-event variability (i.e., comparison among different earthquakes), by considering earthquakes occurred in different seismotectonic and climatic settings.

The categorization of LND and LAP values may be useful to investigate the variable degree of damage on the territory. Xu et al. (2013) propose numerical thresholds to correlate LND and LAP with macroseismic intensity (Chinese scale) following the 2008 Wenchuan earthquake. Hancox et al. (2002) included information on landslides triggered by historical earthquakes in New Zealand for assigning intensities on the Modified Mercalli (MM) scale. Beyond earthquake-induced landslides, Bessette-Kirton et al. (2019) analyzed failures triggered in Puerto Rico (US) by Hurricane Maria using a 2 km x 2 km grid; they classified the territory as either having no landslides, low density (1-25 landslides/km$^2$) and high density (> 25 landslides/km$^2$).

In Fig. 9, the median LND and LAP values derived for the Sabah earthquake are compared to the thresholds proposed by Xu et al. (2013). Both intra- and inter-event variability can be noticed: the application of different area-volume relations results in different estimates of LND and LAP for the Sabah case history; one possible way to handle the epistemic uncertainty due to the existence of different area-volume scaling laws is to include them in a logic tree, where each branch has a weight defined by the modeler. Fig. 9 also shows that thresholds proposed for the Wenchuan earthquake (Chinese intensity scale) are lower than the values obtained for Sabah (ESI-07 scale). One limitation of the data in Fig. 9 is that ESI-07 and Chinese intensity scales are not fully comparable. Inter-event variability is not surprising, and a more comprehensive assessment may be the focus of future efforts: as a research hypothesis, I propose to apply the workflow presented here for the Sabah case to several inventories of earthquake-triggered landslides (Schmitt et al., 2017; Tanyas et al., 2017). The ESI-07 scale seems the most appropriate classification, since it is based only on earthquake environmental effects, and it has a global validity. A statistical approach can then be pursued, investigating either the intra-event (e.g., dispersion of LND and LAP values for each intensity class) and inter-event (comparison among different earthquakes) variability. Geostatistical models (e.g., Lombardo et al., 2021) could be applied as well.

The methodological workflow presented here can be applied to other case histories, to obtain more reliable scaling relations among LND/LAP and ESI-07, eventually tuned according to climatic or seismological parameters, or to the type of slope movement or hillslope material. One way to measure the impact of the methodological workflow presented in this research is its eventual implementation into near real-time products. Currently, institutions such as USGS produce Shakemaps and ground failure estimates in the immediate aftermath of strong earthquakes. These provide information on expected earthquake effects using different descriptors. Maps are expressed in terms of intensity (Modified Mercalli scale) or ground motions (PGA, peak ground acceleration or PGV, peak ground velocity); maps of expected environmental effects (landslides and liquefaction) are produced as well. A similar map expressed in terms of ESI-07 intensity could be an added value with respect to the extant practice.

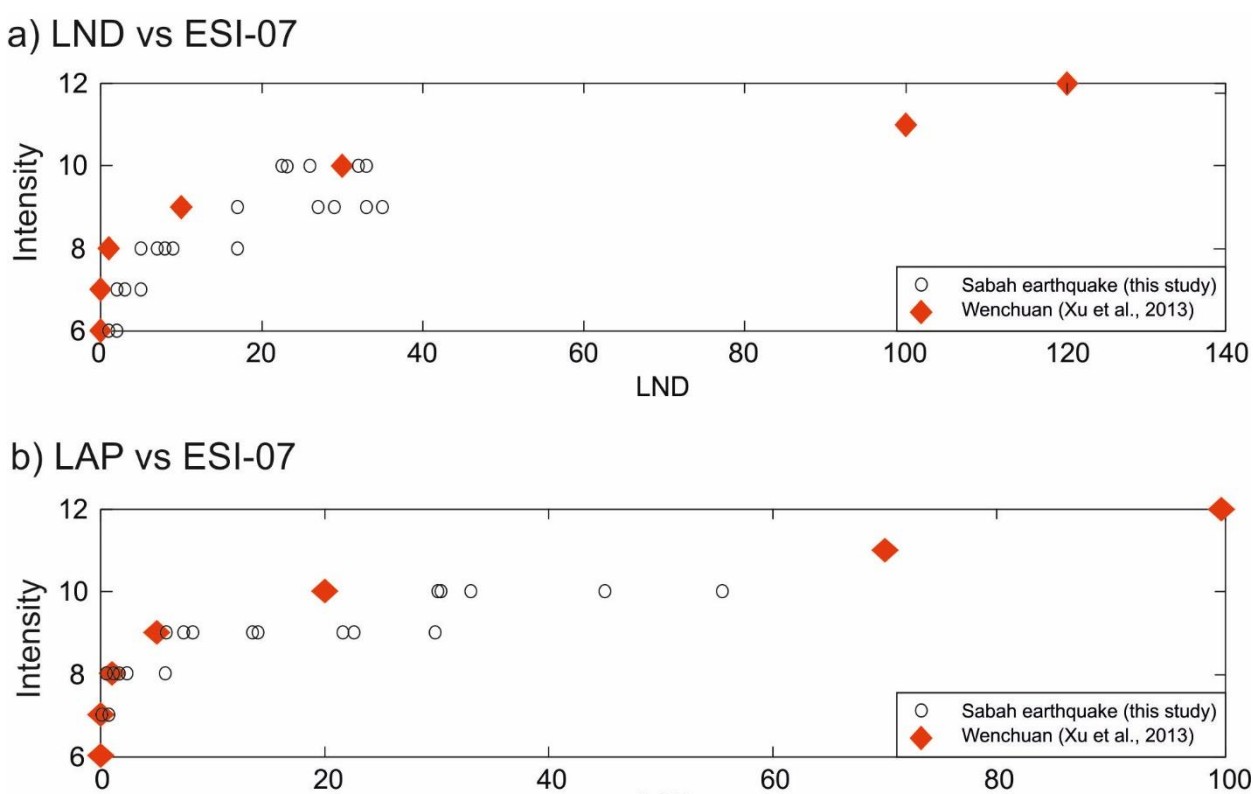

**Figure 9: Plots of LND (a) and LAP (b) vs ESI-07 values; small black circles are the median values for the Sabah earthquake, obtained with the different scaling laws. Red diamonds are the values proposed by Xu et al. (2013) for the 2008 Wenchuan earthquake. This study adopts the ESI-07 scale, while the classes by Xu et al. (2013) refer to the Chinese intensity scale.**

**6 Conclusions**

In this paper, I present an inventory of 5198 landslides triggered by the Mw 6.0 Sabah earthquake, occurred on 4 June 2015. I investigate the spatial pattern of landslides by means of the Landslide Number Density (LND) and the Landslide Area Percentage (LAP) on a regular grid of 1-km² cells. I estimate the ESI-07 intensity for each cell taking advantage of published area-volume scaling relations and demonstrating that the epistemic uncertainty related to the chosen equation has limited

implications on the final output.

I compare the Sabah earthquake with other events on a global scale, finding a good correspondence in terms of total affected area and ESI-07 intensity. I believe that the methodological workflow presented in this paper can be successfully exported in other territorial settings and that joining scientific communities that rarely share their results (e.g., communities responsible for the realization of inventories and for ESI-07 scale assessment) is beneficial for a more comprehensive understanding of

the overall earthquake damage.

Data availability: The inventory in shapefile format and data used to draw Fig. 7 are available on the Zenodo repository (https://zenodo.org/record/6107187#.Yg0AHZbSI2w). Plate boundaries (Fig 1a) are from

https://github.com/fraxen/tectonicplates, earthquakes are from the USGS catalogue (https://earthquake.usgs.gov/earthquakes/search/). The USGS page (USGS, 2018) for the Sabah earthquake is available at https://earthquake.usgs.gov/earthquakes/eventpage/us20002m5s/executive. ALOS-DEM AW3D30 is provided by JAXA (Japan Aerospace Exploration Agency) and is available at https://www.eorc.jaxa.jp/ALOS/en/dataset/aw3d30/aw3d30_e.htm.


Acknowledgments: Planet is thanked for providing PlanetScope imagery as part of the Education and Research program (https://www.planet.com/). I chose the Sabah earthquake as a case history following the post appeared on "The landslide blog" by Dave Petley (https://blogs.agu.org/landslideblog/2019/12/15/mount-kinabalu-google-earth/). Six students from Liceo Galilei (Erba, Italy) helped in realizing the inventory in the framework of a PCTO project.


Declaration of competing interests: The authors declare that they have no conflict of interest.

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
