# Peer review of "Landslides triggered by the 2015 Mw 6.0 Sabah (Malaysia) earthquake: inventory and ESI-07 intensity assignment"

_Natural Hazards and Earth System Sciences, 2022_

## Author Comment (AC2)

**Landslides triggered by the 2015 Mw 6.0 Sabah (Malaysia) earthquake: inventory and ESI-07 intensity assignment**

**Response to Reviewer 2**

I wish to thank Vipin Kumar for the thoughtful comments, which helped in improve the quality of the manuscript. Here I provide a point-to-point answer to all the comments raised by reviewer 2. Original comments are shown in *italic*, while my answer is in plain text.

Following the editorial comments, I added the copyright icon in Figure 2c and I now use the color scheme of Crameri et al. (2020).

**RC2**: 'Comment on nhess-2022-69', Vipin Kumar, 30 Apr 2022 reply

**Comments on the research article (NHESS-2022-69):**

Author aims to elaborate spatial patterns of earthquake (2015 Mw 6.0 Sabah (Malaysia) triggered landslides using Landslide Number Density (LND) and Landslide Area Percentage (LAP). Author further applies the Environmental Seismic Intensity (ESI) scale-2007, considering epicentral intensity of IX (based on landslide affected area). The ESI-07 is used by utilizing the volume of landslides, which is determined using published landslide Area-Volume scaling relationships. The article is mostly well written except few clarifications/elaborations that will make potential readers having diverse backgrounds more interested.

**Comments**:

Author needs to include both pre-earthquake landslide (if any) and post-earthquake landslide inventory of the study area to effectively demarcate the "co-seismic landslide affected area". This affected area is crucial in view of the utilization of ESI-07 scale.

Agreed. In the original paper, I did not focus on pre-existing landslides, since from a visual overview of the satellite images they seemed negligible.

Following the reviewer suggestion, I now realize an inventory of landslides that were already existing before the earthquake occurred on 4 June 2015. While the co-seismic landslide inventory was realized on a homogeneous dataset of satellite images provided by PlanetScope, similar images are not available for the timeframe antecedent the earthquake. Thus, I used Google Earth historical images, ranging from 19 May 2008 to 2 June 2015; this process results in a dataset much more heterogeneous

with respect to the coseismic inventory. Additionally, cloud-free images are not available for 186 km$^2$, corresponding to 23% of the total area (namely, 810 km$^2$-wide).

Figure 1 shows the position of pre-existing landslides with respect to the study area. Inside the pale blue area, no cloud-free images are available. The inventory contains 225 landslides, having a total area of 0.55 km$^2$. For the sake of comparison, the coseismic inventory described in the paper contains 5198 landslides, having a total area of 18.84 km$^2$.

[Figure]

*Figure 1: study area (black polygon) and inventory of landslides mapped on images acquired May 2008 and June 2015. The pale blue area represents a region where cloud-free pre-earthquake imagery is lacking.*

Figure 2 shows an example of multi-temporal images, acquired between 2013 and 2018. The oldest image was acquired in July 2013 and shows a slope movement at the bottom right corner; the second image was acquired one year later (July 2014) and shows more widespread landslides.

The last image was acquired in August 2018 (i.e., after the 2015 Sabah earthquake); it can be noticed that some of the former landslides are no more clearly recognizable in the image.

[Figure]

a) image taken on 27/07/2013

b) image taken on 26/07/2014

c) image taken on 20/08/2018

© Google Earth

*Figure 2: multi-temporal images, showing the evolution of landslides through time. Location is shown in Figure 1.*

*Author also needs to recalculate the LND and LAP in view of the possible changes in the inventory caused by exclusion of pre-earthquake landslides.*

The pre-earthquake landslide inventory, described in the previous response, is significantly smaller than the co-seismic inventory: the former includes 225 landslides, while following the earthquake I mapped 5198 landslides.

In Figure 3, I provide a map of LND and LAP referring solely on the pre-earthquake landslides and I compare them with the coseismic landslides. It is evident the very limited role of preexisting landslides, either in terms of Landslide Number Density (LND) and of Landslide Area Percentage (LAP); more than 95% of the grid cells belong to the lowest class (LND 0 – 1, Figure 3a), while more than 99% of the cells belong to the lowest LAP class (Figure 3b; max LAP value is 0.09%).

**LANDSLIDE INVENTORY PRE-EARTHQUAKE**

**a) Landslide Number Density (LND)**

[Figure]

**b) Landslide Area Percentage (LAP)**

[Figure]

**COSEISMIC INVENTORY**

**c) Landslide Number Density (LND)    d) Landslide Area Percentage (LAP)**

[Figure]

*Figure 3: comparison between LND and LAP computed on the landslide inventory preceding the earthquake (a, b) and on the post-seismic inventory (c, d).*

Following the above reasoning, I did not include the landslide inventory preceding the earthquake, mainly because it is less reliable from the methodological point of view (see above the response to the first comment).

In the manuscript, I added in Section 4.1 a few lines to clarify the negligible influence of pre-existing landslides on the ESI-07 intensity assignment.

*Though it might be difficult to classify 5198 landslides based on type, effective usage of the landslide area-volume (A-V) scaling relationships require type classification. Most of these A-V*

*scaling relationships have been obtained in specific geological and/or climatic settings and have been subjected to defined hillslope material. Notably, Larsen et al. (2010), who used an inventory of >4000 landslides, observed that γ varies based on hillslope material. Further, why did the author include only 6 of the many published landslide A-V scaling relationships?*

In the Sabah inventory, I did not attempt to classify the slope movements according to their type, since my mapping is based on satellite images only. Indeed, such classification would have required detailed field data. It must be underlined that only a small part of the available earthquake-induced landslide inventories includes a genetic description or a classification of the type of movement. For the purposes of the current study, i.e., classification of the earthquake effects using the ESI-07 scale, the input data is the volume of individual slope movements.

In the revised version I consider 8 A-V relations instead of 6 (see next response); I based the selection of these relations among the many published ones on some (rather subjective) criteria, favoring those with global validity (Guzzetti et al., 2009; Larsen et al., 2010) or explicitly related to earthquake-triggered movements (e.g., Xu et al., 2016; Massey et al., 2020).

*Earthquakes generally result in many rock fall type landslides, as the author also showed in Fig. 2c. Author could have included some A-V relationships that have been proposed for rock falls.*

In the revised version of the paper, I consider 8 A-V relation; I added 2 relations specifically dealing with rockfalls. Benjamin et al. (2018) investigated rockfalls on coastal cliffs at Staithes (UK); they compute the volumes adopting 2D and 3D change detection algorithm from Terrestrial Laser Scanner point clouds. In the paper, I consider their 2D equation, since I believe this method is more similar to the other considered relations and can be more widely applied. Caputo et al. (2018) studied rockfalls on coastal cliffs at Coroglio (S Italy), estimating the A-V relations from Terrestrial Laser Scanner data.

One limitation of the Benjamin et al. (2018) and Caputo et al. (2018) relations is the dimension of the individual rockfalls: in the former study, the biggest rockfall is of 27 $m^3$, while in the second study the samples have an area of $0.1 - 10$ $m^2$. The extrapolation of the A-V relations to much bigger landslides should be carefully considered; nevertheless, the application of the 8 relations considered in this study clearly show a similar picture in terms of ESI-07 distribution, testifying that input data (i.e., landslide inventory) is far more important than the choice of the A-V relation.

---

## Author Response (AR1)

**Landslides triggered by the 2015 Mw 6.0 Sabah (Malaysia) earthquake: inventory and ESI-07 intensity assignment**

**Response to Editor**

I wish to thank the editor and reviewers for their thoughtful comments, which helped in improve the quality of the manuscript. Here I provide a point-to-point answer to all the comments. Original comments are shown in *italic*, while my answer is in plain text.

Following the editorial comments, I added the copyright icon in Figure 2c and I now use the color scheme of Crameri et al. (2020).

*Following the comments from the reviewers, the editor suggests to reconsider the paper after major revisions. In particular, the author should add more details on the revised version of the manuscript on the following topics:*

*1- Landslide inventory:*

*Impact of the 8 months delay between the earthquake and the mapping of the landslides? What are the associated uncertainties?*

I agree that this is a critical point that have to be better addressed in the paper. In the discussion, I added a paragraph (Section 5.1) specifically dealing with the challenges in landslide mapping, including the 8 months delay between the earthquake and the image acquisition. The new text is as follows:

Several processes can trigger landslides, such as seismic shaking, heavy rainfall, anthropic disturbances. These processes may act concurrently or have complex interdependencies among each other, resulting in a so-called chain of hazards. The identification of the precise event that triggered the landslides can be challenging, and subsequent remobilizations may occur as well. In the Sabah case, slope movements were firstly triggered by the seismic shaking; later on, prolonged rainfall reactivated the landslide deposits as debris flows (Rosli et al., 2021). In this work, landslides were mapped on optical satellite images, whose availability depends on satellite revisit time and local weather conditions. The co-seismic landslide inventory is realized on images acquired about 8 months after the Sabah earthquake, since persistent cloud cover hampered the analysis of a shorter time interval. This point is a significant source of epistemic uncertainty, which is difficult to reduce, unless other data sources are present (e.g., field surveys, helicopter/drone flights).

Chain of hazards affects the territory for prolonged times: the remobilization of deposits results in enhanced rates of slope movements; these processes may take 5-10 years (Avsar et al., 2016) and generate bank erosion or floodplain accretion downstream, thus affecting flood frequency (Fan et al., 2019). Stochastic natural processes (e.g., earthquakes) and seasonal hazards (e.g., rainfall, flood) imply different modeling tools and calls for complex risk reduction strategies (Quigley et al., 2020); understanding cause-effect relationships and latent vulnerabilities helps in informing such efforts (Pescaroli & Alexander, 2016). Additionally, landslide phenomena triggered by human activities are increasing and have an influence comparable, if not higher, than natural processes such as rainfall or earthquakes (Froude & Petley, 2018; Tanyas et al., 2022).

*What about re-vegetation of the areas?*

I think that re-vegetation does not affect the possibility to map the landslides. The study area is covered by thick forest; landslides are easily recognizable since they completely stripe off the vegetation cover. Figure 1 shows satellite images acquired at different times; panel a) is pre-earthquake; panel b) is acquired in March 2016 and is a sample of the imagery used to draw the landslide inventory; panel c) shows the same spot 2 years later, when landslides are still clearly identifiable.

[Figure]

*Figure 1: multi-temporal images, showing the evolution of landslides through time; a) Google Earth image taken on 29/09/2008 (before the earthquake); b) PlanetScope image taken on 18/03/2016 (10 months after the earthquake); c) PlanetScope image taken on 9/03/2018 (ca. 3 years after the earthquake).*

*How are coalescent landslides mapped? The use of the « split » tool (line 125) should be explained in more details.*

Text modified as follows:

Landslide inventories may suffer from problems related to amalgamation of coalescing polygons, i.e., the mapping of several adjacent landslides as a single polygon (Marc and Hovius, 2015). This

problem is especially severe for inventories realized through automatic mapping and may introduce a bias in the computation of landslide number and other statistics (e.g., ESI-07 assessment). When multiple sources areas coalesce in a single toe sector, it is difficult to identify individual landslides. In such cases, I first mapped the entire polygon, then I used the "split" tool to delineate the different source areas. This GIS tool allows to draw contiguous polygons, avoiding the overlap of different polygons, or unmapped areas in between.

*Pre- and post-earthquake landslides inventories: the influence of pre-earthquake landslide inventory has to be shown explicitely as suggested by reviewer 2.*

*Comments provided by the author on the supplementary material to questions from reviewer 2 could serve as a basis to improve this topic on the manuscript.*

Agreed. In the original paper, I did not focus on pre-existing landslides, since from a visual overview of the satellite images they seemed negligible.

Following the reviewer suggestion, I now realize an inventory of landslides that were already existing before the earthquake occurred on 4 June 2015; the role of pre-existing landslides is discussed in Section 5.1.1, where I also added a new image showing LND and LAP for the pre-existing landslides. The new text is as follows:

A co-seismic landslide inventory should include only those slope movements triggered, or reactivated, by the seismic shaking. The inventory presented in this paper is realized on a homogeneous dataset of satellite images provided by PlanetScope; similar images are not available for the timeframe antecedent the earthquake, introducing a difficulty in the evaluation of whether a landslide was already present before the earthquake. Thus, I realized a dataset of pre-existing landslides by inspecting Google Earth historical images, ranging from 19 May 2008 to 2 June 2015. Cloud-free images are not available for a 186 km$^2$ wide region, corresponding to 23% of the total area (blue region in Figure 6). I mapped a total of 225 pre-existing landslides and compute LND and LAP with the same procedure adopted for the co-seismic inventory. It is evident that pre-existing landslides exert a very limited role, either in terms of total number (225 pre-existing vs 5198 co-seismic) and area (0.55 vs 18.84 km$^2$). In Figure 6, I adopted the same color scheme as for the co-seismic inventory (Figure 4), to highlight that more than 95% of the grid cells belong to the lowest LND class (less than 2 landslides per cell), while more than 99% of the cells belong to the lowest LAP class (max LAP value is 0.09%).

[Figure]

*Figure 2: Grid maps of Landslide Number Density (LND, a) and Landslide Area Percentage (LAP, b) for pre-existing landslides; the blue area represents a region where cloud-free pre-earthquake imagery is lacking. Colormap follows Crameri et al. (2020).*

*2- area-volume scaling laws: As highlighted by reviewer 2, are the area-volume scaling laws all valid for the types of movements included in the current landslide inventory? This should be checked and shown in the paper. Besides, are the area-volume scaling laws all valid for landslides identified by their « source area + deposit »?*

Thanks for the comment, this is indeed a point I should have considered more cautiously since the early stages.

In the revised version I consider 7 A-V relations; I based the selection of these relations among the many published ones on some (rather subjective) criteria, favoring those with global validity (Guzzetti et al., 2009; Larsen et al., 2010) or explicitly related to earthquake-triggered movements (e.g., Xu et al., 2016). I deleted the Massey et al. (2020) equation, since it is valid for "source only" polygons. I checked the original papers and the equations by Guzzetti et al. (2009), Larsen et al. (2010) and Xu et al. (2016) are valid for "source area + deposit" polygons, so I retain them in the paper.

I added 2 relations specifically dealing with rockfalls. Benjamin et al. (2018) investigated rockfalls on coastal cliffs at Staithes (UK); they compute the volumes adopting 2D and 3D change detection algorithm from Terrestrial Laser Scanner point clouds. In the paper, I consider their 2D equation, since I believe this method is more similar to the other considered relations and can be more widely

applied. Caputo et al. (2018) studied rockfalls on coastal cliffs at Coroglio (S Italy), estimating the A-V relations from Terrestrial Laser Scanner data.

*The authors should further comment on their choices of the 6 laws and add one for rockfalls as suggested by reviewer 2. Comments provided by the author on the supplementary material to questions from reviewer 2 could serve as a basis to improve this topic on the manuscript.*

In the Sabah inventory, I did not attempt to classify the slope movements according to their type, since my mapping is based on satellite images only. Indeed, such classification would have required detailed field data. It must be underlined that only a small part of the available earthquake-induced landslide inventories includes a genetic description or a classification of the type of movement. For the purposes of the current study, i.e., classification of the earthquake effects using the ESI-07 scale, the input data is the volume of individual slope movements.

One limitation of the equations specific for rockfalls (Benjamin et al., 2018 and Caputo et al., 2018) is the dimension of the individual rockfalls, which in both cases do not exceed 30 m2. The extrapolation of the A-V relations to much bigger landslides should be carefully considered; nevertheless, the 7 relations considered in this study clearly show a similar picture in terms of ESI-07 distribution, testifying that input data (i.e., landslide inventory) are far more important than the choice of the area-volume relation.

In Section 5.1.2 I further discuss the implications of the selection of the A-V relation, the new text is as follows:

Another source of epistemic uncertainty is related to the area-volume scaling relations adopted to compute ESI-07 values. Many equations have been proposed in the literature, referring to different triggering processes (e.g., seismic shaking vs rainfall), climate conditions (specific regions vs global validity), landslide type (slides vs rockfall), mapping procedures (e.g., landslides delineated as single polygons vs separation of source and deposit area) and methods for data acquisition (e.g., manual vs automatic mapping; satellite vs drone images vs laser scanner techniques). Thus, the selection of the most suitable equation may not be straightforward. In Section 4.2 I used seven different equations to derive the ESI-07 macroseismic field; results demonstrate that the epistemic uncertainty related to the choice of the area-volume relation is much lower than other sources of uncertainty.

*3- Quantitative relation between LAP / LND and ESI-07 (line 210): according to this editor, it is not clearly outlined in the paper. Besides, isn't the correlation between LAP and ESI-07 expected*

*because both parameters depend on landslide area?*

Thanks for the comment. I address this point in two ways:

- I added a new figure in the methods section, to highlight the role of number and dimension of landslides on the computation of LND, LAP and ESI-07.
- I moved the section on scaling relations between LND, LAP and ESI-07 toward the end of the paper, to answer the request for a clearer distinction between results and discussion, and to introduce the final section of the discussion, dealing with the prospect for future work.

Text added in the methods section:

LND, LAP and ESI-07 focus on different aspects, as illustrated in Figure 3. The three panels have the same LAP (36% of the "study region", i.e., the black square); the area of the biggest landslide is used to compute the ESI-07 value, as done for the real case study. In this example, the Guzzetti et al. (2009) equation is used to illustrate the results. The first scenario (Figure 3a) shows one wide landslide, resulting in an ESI-07 value ≥X. The second case shows 36 small landslides and is equivalent to ESI-07 VIII. The third case shows the presence of one medium-sized and 30 small landslides, resulting in ESI-07 value of IX. Figure 3 highlights that the concurrent evaluation of LND, LAP and ESI-07 provide an added value in the understanding of the distribution and characteristics of the landslide inventory, due to the role played by the number and dimension of individual landslides in the calculation of the different metrics.

[Figure]

| | A | B | C |
|---|---|---|---|
| Number of landslides | 1 | 36 | 31 |
| Dimension biggest landslide (m) | 600 x 600 | 100 x 100 | 300 x 200 |
| Area biggest landslide (m2) | 360000 | 10000 | 60000 |
| Cumulative area (m2) | 360000 | 360000 | 360000 |
| Landslide area percentage (LAP) | 0.36 | 0.36 | 0.36 |
| **ESI-07 intensity** | **>X** | **VIII** | **IX** |

*Figure 3: Sketch illustrating the role of number and dimension of landslides (coloured squares) on the computation of LND, LAP and ESI-07. The upper panels represent simplified scenarios: a) one wide landslide; b) many landslides, all of small dimension; c) many landslides with variable dimension.*

*4- Comparison to other case studies: is the Wenchuan landslide inventory a good candidate for this comparison. To this editor, it appears that this event has specific characteristics that do not mimic most of the earthquake-triggered landslides inventories worldwide. Therefore, work presented in particular in Fig. 6 raises questions. Besides, what is the purpose of comparing two very different intensity scales (larger densities of landslides in Wenchuan require additional classes in the chinese intensity scale)?*

I agree that the Wenchuan earthquake may not be representative of available landslide inventories. I moved the comparison with the Wenchuan earthquake in the last section of the discussion, entitled "prospect for future work". Indeed, even though the Wenchuan earthquake may not be optimal for a comparison, it is the only case where an attempt has been made to compare LND/LAP with an intensity scale. In the revised text, less emphasis is put on the comparison Sabah vs Wenchuan earthquake, since this is presented as an avenue for future research.

The modified text is as follows:

as a research hypothesis, I propose to apply the workflow presented here for the Sabah case to several inventories of earthquake-triggered landslides (Schmitt et al., 2017; Tanyas et al., 2017).

The ESI-07 scale seems the most appropriate classification, since it is based only on earthquake environmental effects, and it has a global validity. A statistical approach can then be pursued, investigating either the intra-event (e.g., dispersion of LND and LAP values for each intensity class) and inter-event (comparison among different earthquakes) variability. Geostatistical models (e.g., Lombardo et al., 2021) could be applied as well.

The methodological workflow presented here can be applied to other case histories, to obtain more reliable scaling relations among LND/LAP and ESI-07, eventually tuned according to climatic or seismological parameters, or to the type of slope movement or hillslope material. One way to measure the impact of the methodological workflow presented in this research is its eventual implementation into near real-time products. Currently, Shakemaps and ground failure models are routinely produced following strong earthquakes; these maps provide information on the expected ground motion (expressed in terms of Modified Mercalli intensity, PGA or PGV) and environmental effects (landslides and liquefaction). A similar map expressed in terms of ESI-07 intensity could be provide an added value with respect to the extant practice.

*5- Discussion: As suggested by reviewer 1, the quality of the paper would be enhanced if the author separates results from discussion, adding a discussion section focusing on more aspects, such as data, methods, results, and comparing with previous work to analyze the advantages and disadvantages of this work. In addition, there is a need to have a prospect for future work. Comments provided by the author on the supplementary material to questions from reviewer 1 could serve as a basis to improve this topic on the manuscript.*

I re-organized the second part of the paper; now results and discussion are two separate sections. Please see previous responses for more details on the single aspects. The new outline of Section 4 (results) comprises two sub-sections, describing the spatial distribution of the mapped landslides and the ESI-07 macroseismic field.

Section 5 (Discussion) is organized as follows:

- In section 5.1 I address the challenges in the landslide mapping, with the issues concerning input data and the methodological steps, such as the role of pre-existing landslides and the date of acquisition of satellite images (8 months after the earthquake). Here I briefly touch the main sources of uncertainties and data limitations; I also added a new figure, i.e., the grid map of LND and LAP for the pre-existing landslides.
- Section 5.2 deals with the comparison of the Sabah case histories with other events worldwide; this part was already included in the original manuscript and allows to move from the single case study to a broader view.

- In Section 5.3 I describe the scaling relations between LND, LAP and ESI-07. This part was formerly in the Results section, but I acknowledge that it is better to place it in the discussion, since this represents the main novelty of my approach.

- Following the reviewers and editor comments, I added a new Section (5.4), delineating the prospect for future work. I moved here the comparison with the Wenchuan landslide inventory, since this topic should be the ground on which a wider study can be realized, investigating all the existing earthquake-induced landslide inventories, to assess inter-event variability.

*Non-public comments to the Author:*

*Others:*

*- Which GIS platform is used in this work (line 112)?*

It is QGIS, sentence modified.

*- Please specify that magnitude (Mw) is used in equations (2), (3).*

Done.

---

## Author Response (AR2)

**Landslides triggered by the 2015 Mw 6.0 Sabah (Malaysia) earthquake: inventory and ESI-07 intensity assignment**

**Response to Editor**

I wish to thank the editor for the thoughtful comments, which helped in improve the quality of the manuscript. Here I provide a point-to-point answer to all the comments. Original comments are shown in *italic*, while my answer is in plain text.

*Comments to the author:*
*The editor suggests that the author make the additional following corrections before publication :*
*- Line 25 : distinction between « primary effects » and « ground shaking » ?*
Primary effects include surface faulting and the permanent deformation of the topographic surface; with "ground shaking" I mean all those effects on the natural and built environment that are due to the passage of seismic waves. New text is as follows:
Moderate to strong earthquakes cause widespread damage due to primary effects (i.e., those related to the seismogenic source, which include surface faulting and permanent ground deformation) or due to ground shaking (i.e., related to the passage of seismic waves).

*- Lines 187 and 190 : fig.4 instead of fig. 3*
Done.

*- Lines 193-194 : could the author add a plot showing the comparison between predictions of landslides occurrence from USGS model and the true distribution of failures in the paper or in the supplementary material to support this statement ?*
Agreed. I added as supplementary material a map showing the expected distribution of landslides according to the USGS model and the actual distribution. The map is appended below.

[Figure]

[Figure]

Figure S1: a) Expected landslide distribution, extracted from USGS website; b) actual distribution of mapped landslides.

- *In LAP, fig.4, add %*

Done.

- *ESI-07 degrees are based on landslide volumes categories defined by ranges of 1 order of magnitude span. Changes in the AV laws does not imply a change in the resulting volumes by one order of magnitude. That may explain why modifying the AV law does not change the final distribution of ESI-07 local intensity (cf. maps of Fig.5). Observed differences between the ESI-07 maps associated to the different laws are related to landslides whose volumes are close to the upper and lower boundaries of the categories defined by the ESI-07 degree scale. Therefore the comment of lines 215-216 should be modified accordingly.*

Agreed, I added the following sentence:

This is because ESI-07 degrees are based on broad categories in terms of volume (each category span at least one order of magnitude, Table 2); observed differences between the ESI-07 maps are related to landslides whose volumes are close to the boundaries defined in the ESI-07 scale.

- *Fig 5 : complete the legend to describe what plots a to g show. Same for plot h.*

New caption is as follows:

Figure 5: Grid maps of ESI-07 local intensity obtained by adopting different area-volume scaling relations (a: Guzzetti et al., 2009; b: Larsen et al., 2010 – all types; c: Larsen et al., 2010 – bedrock; d: Larsen et al., 2010 – soil; e: Xu et al., 2016; f: Benjamin et al., 2018; g: Caputo et al., 2018); h: frequency of cells belonging to the different ESI-07 classes for each scaling law.

*- Lines 305-306 : as mentionned by the author, we can not exclude that the higher number of landslides in the Sabah earthquake is only due to the fact that the inventory was realized 8 months later, therefore incorporating also remobilizations. This leads to comparisons with other earthquakes being treated with caution.*

Agreed, new text is as follows:

I realized the inventory on satellite images acquired 8 months after the earthquake, thus slope movements triggered by processes other than the mainshock may be included, such as debris flow remobilization. This implies that comparison with other earthquakes should be gingerly considered.

*- Table 4 : are LAP values given in this table in % ? If so, please add « % » in the legend. Are more than 2 digits meaningful ?*

Agreed, and corrected accordingly.

*- Line 314 : LAP may better represent ESI-07 because ESI-07 is based on volume and volume on area, right ?*

I expect generally to be true, but still not necessarily. A similar LAP may be obtained by many small landslides, or by one bigger landslide. ESI-07 depends on the biggest landslide in a cell unit. For instance, Figures 3a and 3b both have LAP = 0.36, but different ESI-07 values. The new text is as follows:

Median LAP values instead do not show such inversions, eventually suggesting that LAP is a better descriptor than LND for assessing the damage. This fact is not surprising, since LND has a "point" validity, while LAP is related to an area assessment, which should generally be more consistent with volume (on which ESI-07 values are based).

*- Line 360 : the meaning of this sentence is not clear to the editor.*

The text has been rephrased as follows:

Currently, institutions such as USGS produce Shakemaps and ground failure estimates in the immediate aftermath of strong earthquakes. These provide information on the expected earthquake

effects using different descriptors. Maps are expressed in terms of intensity (Modified Mercalli scale) or ground motions (PGA, peak ground acceleration and PGV, peak ground velocity); maps of expected environmental effects (landslides and liquefaction) are produced as well.

*- Fig. 9 : it should be explained into more details : how were selected the boundaries of each ESI-07 scale for LND parameter for instance ? How can we tell looking at Fig. 8 ? The median value is only one parameter : therefore is it used to define the lower or the upper boundary of each ESI-07 class ?*

Thanks for the comment. I realized that the type of graph I selected was conveying unclear information. I changed the graph style, making it similar to Figure 8. Points represent the median values; I believe the new graph is more consistent with the preliminary nature of the data currently available. The analysis of more case studies will allow to derive class boundaries of each ESI-07 degree on sound statistical basis.

In the text, I changed the description referring to Fig. 9; the new version of the figure is appended below. New text:

In Fig. 9, the median LND and LAP values derived for the Sabah earthquake are compared to the thresholds proposed by Xu et al. (2013). Both intra- and inter-event variability can be noticed: the application of different area-volume relations results in different estimates of LND and LAP for the Sabah case history; one possible way to handle the epistemic uncertainty due to the existence of different area-volume scaling laws is to include them in a logic tree, where each branch has a weight defined by the modeler. Fig. 9 also shows that thresholds proposed for the Wenchuan earthquake (Chinese intensity scales) are lower than the values obtained for Sabah (ESI-07 scale).

[Figure]

Figure 9: Plots of LND (a) and LAP (b) vs ESI-07 values; small black circles are the median values for the Sabah earthquake, obtained with the different scaling laws. Red diamonds are the values proposed by Xu et al. (2013) for the 2008 Wenchuan earthquake.